# Enhancement of Thermal Tolerance and Growth Performances of Asian Seabass (*Lates calcarifer*) Fed with Grape Extract Supplemented Feed

**DOI:** 10.3390/ani14182731

**Published:** 2024-09-20

**Authors:** Salman Akram, Naveen Ranasinghe, Tsung-Han Lee, Chi-Chung Chou

**Affiliations:** 1Department of Veterinary Medicine, College of Veterinary Medicine, National Chung Hsing University, Taichung 402, Taiwan; d110052010@mail.nchu.edu.tw; 2Department of Life Sciences, National Chung Hsing University, Taichung 402, Taiwan; madhumalranasinghe@gmail.com (N.R.); thlee@email.nchu.edu.tw (T.-H.L.); 3The iEGG and Animal Biotechnology Center, National Chung Hsing University, Taichung 402, Taiwan

**Keywords:** Asian seabass, grape extract, antioxidant enzymes, cold tolerance

## Abstract

**Simple Summary:**

Cold snaps during winter pose a significant challenge for Asian seabass (*Lates calcarifer*) in Taiwan. Such temperature stress can reduce antioxidant enzyme activity, burdening these enzymes and requiring additional support from non-enzymatic antioxidants. Grape extract (GE), rich in polyphenols, may enhance the antioxidant system in these fish. In a 60-day feeding experiment, various levels of GE (20, 30, and 40 g per kilogram of feed) were tested against a control group to assess their impact on growth, antioxidant-related gene expression, and cold tolerance. The findings demonstrated that the group receiving 20 g of GE per kilogram of feed (GE20) experienced higher improvements in weight gain, antioxidant gene expression, and cold tolerance, suggesting that GE at this concentration could serve as an effective dietary supplement to help Asian seabass withstand the stresses of cold temperatures during cold snaps.

**Abstract:**

Cold snaps during the winter present a critical challenge for Asian seabass (*Lates calcarifer*) in Taiwan, as sudden temperature drops significantly affect their growth and survival. This study explores the effects of dietary grape extract (GE) from *Vitis vinifera* on the growth performance, oxidative stress regulation, and thermal tolerance of this commercially valuable fish. Over a 60-day feeding trial, four dietary groups were tested: a control diet without GE and three diets supplemented with GE at 2% (GE20), 3% (GE30), and 4% (GE40) with commercial feed. The results demonstrated that GE supplementation positively influenced growth, with the GE20 group achieving the best weight gain and feed conversion ratio among all groups. The upregulation of the growth-related gene *igf-1* in the liver of the GE20 group further supported its superior growth performance. Additionally, GE-fed groups showed increased expression of antioxidant-related genes *sod1* and *sod2* in the liver, while *gpx1* exhibited a significant increase only in the GE20 group, indicating enhanced antioxidant defenses. *Cat* gene expression remained unchanged, and higher GE doses reduced the expression of *gpx1*, *cat*, and *igf-1*. Furthermore, GE supplementation improved cold tolerance in all treated groups compared to the control. These findings suggest that dietary GE at 20 g/kg is particularly effective in enhancing growth performance and cold tolerance in Asian seabass, offering a promising strategy for boosting fish health and adaptability in aquaculture.

## 1. Introduction

The incorporation of bioactive compounds, such as polyphenols, into standard aquaculture diets is increasingly recognized as a strategy to enhance the overall health and productivity of fish species. Modern aquaculture practices emphasize the importance of balanced nutrition for promoting optimal growth and improving the health and resilience of aquatic organisms [1]. Traditionally, aquaculture diets have been formulated with essential components like proteins, carbohydrates, lipids, vitamins, and minerals. However, recent advancements highlight the potential of supplementing these diets with additional bioactive compounds, particularly those derived from natural sources [2]. Polyphenols have garnered significant attention due to their multifaceted health benefits. These naturally occurring compounds, abundant in various plants, are known for their potent antioxidant, anti-inflammatory, and antimicrobial properties [3]. Phenolic compounds effectively neutralize reactive oxygen species (ROS), which are harmful molecules capable of damaging cellular structures and functions [4]. Polyphenols, encompassing various classes such as flavonoids, phenolic acids, stilbenes, and lignans, are pivotal in mitigating oxidative stress through multiple mechanisms. Polyphenols, particularly those containing catechol groups (a benzene ring with two hydroxyls), can form complexes with metal ions. They also donate electrons to peroxidases, facilitating the conversion of hydrogen peroxide into water. During oxidation within cells, polyphenols generate metabolites that can stimulate the expression of genes encoding antioxidant enzymes such as superoxide dismutase (*sod*) and glutathione peroxidase (*gpx*) and increase the availability of nicotinamide adenine dinucleotide phosphate (NADPH), which is crucial for the regeneration of antioxidants and cellular defense against oxidative damage. They also activate vital signaling pathways, including PI3 K, Akt, Nrf2, and HO-1, further upregulating the cellular antioxidant response [5].

Numerous scientific studies have specifically focused on examining phenolic compound’s impact on cold-blooded vertebrates such as fish. These creatures possess unique metabolic and physiological traits, making them a distinctive focus of research in this area [6]. Phenolic compounds improved the inflammation response and immunological potential of the zebrafish intestine, suggesting that they could be used as a sustainable feed additive in aquaculture [7,8]. Recently, the chia seeds (*Salvia hispanica*) and rocket (*Eruca sativa*), rich in phenolic compounds, have been reported to alleviate stress and enhance survival, growth, antioxidant, and immunological parameters in tilapia exposed to low temperatures. Their inclusion in aquaculture diets represents a cost-effective approach to enhancing fish health and improving the sustainability of aquaculture practices [9].

Grape (*Vitis vinifera*) is a good nutritional source of polyphenols. It is rich in flavonoids and phenolic compounds, which exhibit powerful antioxidant properties comparable to other known natural compounds like ascorbic acid and calciferol [10]. The consumption of grapes offers various health advantages, such as anti-aging effects, ulcer relief, inflammation reduction, and protection against atherosclerosis, along with its antioxidant and antimicrobial characteristics. The antioxidant effect is achieved by triggering essential antioxidant genes and enhancing both enzymatic and non-enzymatic defense systems in both animals and humans [11,12]. Previous research has shown that incorporating grape seed extract into the diet of rainbow trout significantly enhances their growth performance and mucosal immunity, as indicated by increased epidermis thickness, goblet cell density, villus height, and intraepithelial lymphocytes in the intestine [13,14]. These growth-promoting effects are partly mediated through the enhancement of the insulin-like growth factor (*igf*) signaling system, which includes IGFs, IGF receptors, and IGF-binding proteins [15,16,17]. The *igf-1* gene, in particular, serves as a critical marker for evaluating growth in fish and shellfish, making it a valuable metric in nutritional research to assess the impact of specific nutrients on aquatic species [18].

In addition to promoting growth, grape extract has the potential to enhance fish resilience to environmental stressors, particularly temperature fluctuations. Asian seabass (*Lates calcarifer*), a euryhaline fish native to the Indo-West Pacific region, is highly significant in aquaculture, with production reaching 95,000 metric tons in 2018 across Taiwan, Malaysia, Thailand, Indonesia, and Australia [19]. Taiwan, once the world leader in Asian seabass aquaculture, experienced a resurgence in production by 2005 and produced 20,000 metric tons in 2020, primarily for export to the United States and Australia [20]. However, its production and the fish’s survival ability are influenced when the water temperature drops below 20 °C [21,22], and mortality may increase when the temperature drops below 15 °C [23]. Temperature drops can lead to cellular stresses and impaired physiological functions such as oxidative stress, disruption of ion exchange across the cellular membrane such as calcium, potassium, and sodium due to changes in membrane fluidity and structure, and perturbation of metabolic processes. These changes can potentially induce oxidative stress, an imbalance between the production of ROS and the body’s antioxidant defense mechanisms, which in turn activates the *nrf2*. This regulatory protein is responsible for controlling the expression of genes involved in the antioxidant defense, such as *sod*, *gpx*, and catalase (*cat*). The upregulation of these genes aids cells in their ability to prevent damage from oxidizing agents [24]. These factors can impact the overall health and functionality of the fish [25]. Incorporating functional dietary plants into the feeding regimen of fish is an effective way to enhance their cold tolerance significantly [26,27,28].

Our study aims to investigate the effects of GE derived from the fruit, which contains phenolic compounds, on the growth, oxidative stress-related gene expressions, and cold tolerance of Asian seabass. By integrating molecular and physiological approaches, we seek to enhance our understanding of the underlying mechanisms that could contribute to improving cold tolerance in this economically important fish species. Specifically, we focus on the role of polyphenols found in grape extract, which has been shown to improve metabolism and antioxidant capacity in fish under cold stress by enhancing the expression of key antioxidant genes like *sod1*, *sod2*, *cat*, and *gpx* [26]. This approach not only offers a natural and sustainable way to boost the resilience of Asian seabass but also provides valuable insights for aquaculture and fisheries management [29,30].

## 2. Materials and Methods

### 2.1. Diet Preparation

Grape extract powder, derived from the fruit of *Vitis vinifera*, was obtained from a company (Noah’s ARK, Taipei, Taiwan) and added to the commercial feed (crude protein, 43%; crude lipid, 3%; ash, 16%; crude fiber, 3%; moisture, 11%) at different concentrations: 2%, 3%, and 4% (*w*/*w*). The GE solution was prepared by dissolving 5 g of GE powder in 100 mL of warm water at 40 °C. The resulting solution had a concentration of 5% *w*/*v*, and the solution was carefully applied to commercial fish feed, using a precision spraying technique aimed at ensuring as consistent an application of GE as possible across all pellets. Control feed was prepared with a spray of pure water to resemble the coating process. GE-coated and control feeds were dried in an oven and stored in a 4 °C refrigerator for later use in the feeding trial.

### 2.2. Experimental Design

Asian seabass, approximately 200 fingerlings of the same size were purchased from a local fish farm in Taichung, Taiwan. The fish were quarantined in well-aerated freshwater for a week. Following quarantine, the fish (initial weight: 21.03 ± 1.20 g) were randomly placed into four treatment groups (control and three treatment groups), with each group consisting of 15 individuals per tank and 30 individuals per group (in duplicate). All fish were reared in 500 L^−1^ fiberglass tanks. The tanks were connected to a recirculatory aquaculture system with constant aeration and physical and biological filtration. Fish were fed twice daily at 08:00 h and 16:00 h with the prepared feeds of apparent satiation. Water quality was maintained within the normal physiochemical parameters for 60 days, including water temperature at 28.00 ± 1 °C, dissolved oxygen and pH were 6.96 ± 0.25 mg L^−1^ and 7.49 ± 0.2, respectively, while non-ionized ammonia (0.19 ± 0.02 mg NH_3_ L^−1^) were assessed weekly. Tanks were cleaned and siphoned daily to remove feces. Following the completion of the feeding trial, a critical minimum temperature (CT_min_) trial was performed to evaluate the thermal tolerance of Asian seabass. In the trial, the fish were transferred to an experimental tank set at a temperature of 28 °C, and the temperature was then continuously lowered by using a water cooler at an average rate of 0.21 ± 0.04 °C min^−1^ following the protocol by Ford and Beitinger [31]. Temperature was measured with a calibrated mercury thermometer. The water temperature at which the fish could not maintain their balance or respond to gentle prodding with a glass rod was then recorded as the CT_min_ endpoint. After that, the fish were returned to their acclimation tank at 28 °C and observed for 24 h to assess their survival.

### 2.3. Growth Performance Parameters

Specific Growth Rate (SGR):SGR (%day^−1^) = 100 × (ln [mean final body weight] − ln [mean initial body weight])/time (days)

Weight Gain (WG):WG (%) = 100 × ([mean final body weight − mean initial         body weight]/mean initial body weight)

Feed intake
FI (%/day) = 100 × total amount of feed consumed/[(initial body weight + final body weight)/2]/t

Feed Conversion Ratio (FCR):FCR = dry feed intake (g)/wet weight gain (g)

### 2.4. Liver Sample Preparation

After 60 days of feeding, the fish were anesthetized using MS−222 (40 mg L^−1^) for about 1–2 min until they stopped swimming and settled at the bottom. Once immobilized, the sampling process was initiated by an incision in the abdominal cavity from the cloaca opening, and the liver was dissected immediately. A total of 10 fish were randomly selected from each experimental group, and liver tissue samples (approximately 50 to 70 mg each) were collected. Liver samples were dissected and placed into 600 µL of TRIzol reagent (Roche, Mannheim, Germany), followed by immediate freezing using liquid nitrogen. The samples were then stored at −80 °C until further analysis.

### 2.5. RNA Extraction and Reverse Transcription

A modified RNA extraction method was used, utilizing TRIzol reagent to enhance the efficiency and yield of the RNA isolation [32]. The process commenced with the homogenization of liver tissues in the TRIzol reagent. Subsequent addition of 60 μL 1–bromo–3–chloropropane (BCP) facilitated the phase separation of the sample into three components: the aqueous phase containing RNA, an interphase, and an organic phase. RNA precipitation occurred by introducing isopropanol, and the resulting RNA pellet was washed with 1 mL of 75% ethanol to remove impurities. The purified RNA pellet was then air-dried and dissolved into RNase-free water. The RNA quality and quantity were assessed using a NanoDrop 2000 spectrophotometer (Thermo, Wilmington, DE, USA). A260/A280 ratios were between 1.9 and 2.1. Additionally, RNA integrity was verified through 1% gel electrophoresis. Following that, first-strand cDNA was synthesized using the iScript™ cDNA Synthesis Kit (Bio-Rad, Hercules, CA, USA) following the manufacturer’s instructions and used in real-time PCR (qPCR) to determine the mRNA levels described below.

### 2.6. Quantitative Real-Time PCR

Protocols previously described by Ranasinghe et al. (2022) [33] for qPCR analysis were followed. Briefly, primers for qPCR analysis were designed based on *Lates calcarifer* sequences obtained from the NCBI database (Table 1). Thermal cycling was performed using the Applied Biosystems^®^ Veriti^®^ 96-Well Thermal Cycler from Thermo Fisher Scientific. Gradient temperatures ranging from 58 to 62 °C were systematically tested to optimize primer performance and thermal cycling conditions. The qPCR reaction mixture consisted of 1.6 µL of the dNTP mixture, 2 µL of 10× Ex Taq reaction buffer, 0.1 µL of Ex Taq polymerase (Takara, Shiga, Japan), 1 µL of cDNA (20× dilution), 0.5 µL of each for forward and reverse primer (250 nM), and nucleotide, DNase, and RNase-free water (Protech, Taipei, Taiwan), then bringing the final volume to 20 µL. The PCR product was verified by using 2% agarose gel electrophoresis to confirm the target sequence length. The confirmed PCR product underwent sequence analysis at Tri-I Biotech Company (Taipei, Taiwan). Sequencing results were cross-validated against the *L. calcarifer* entries in the NCBI online database. 

The quantification of mRNA expression levels for *gpx1*, *sod1*, *sod2*, *cat*, and *igf-1* was conducted using the Rotor-Gene Real-Time PCR System (QIAGEN, Hilden, Germany), with expression levels normalized to the housekeeping gene *elf1α*, following these steps: 2 µL of cDNA, primers (0.5 µL) at a concentration of 250 nM (adjusted based on primer efficiency), and 10 µL of KAPA SYBR FAST qPCR Master Mix (KAPA Biosystems, Cape Town, South Africa), with nucleotide, DNase, and RNase-free water (Protech) added to achieve a total volume of 20 µL. Non-specific products were assessed through both melting curve analysis and gel electrophoresis of the primers to ensure specificity. The primer efficiency was maintained within the range of 95–105% as a measure of primer specificity. The mRNA expression levels of the genes in this study were calculated using the 2^−ΔΔCt^ method [34].

### 2.7. Data Analysis

All data in the present study were analyzed using one-way analysis of variance (ANOVA) with the SPSS software package, Version 20.0 (IBM Corp., Armonk, NY, USA). Post hoc comparisons were performed using Duncan’s multiple range test to identify significant differences among the experimental groups. All graphical representations were created using GraphPad Prism Version 9.0 (GraphPad Software, San Diego, CA, USA). Results were presented as mean ± standard error of the mean (SEM), with a significance level set at *p* < 0.05.

## 3. Results

### 3.1. Effects on Growth Performances

The growth performance of Asian seabass subjected to varying amounts of grape extract (GE) is summarized in Table 2. The findings revealed that the group fed with 20 g/kg of GE exhibited the highest growth performance among the three tested groups and was significantly different from the control group in final weight, weight gain, feed conversion ratio (FCR), and specific growth rate percentage (SGR%). The result suggests that 20 g/kg of GE could be an optimal dosage for enhancing the growth of Asian seabass. The expression of growth-related gene *igf-1* is shown in Figure 1. The mRNA level of *igf-1* in the fish liver was significantly higher than the control and other groups (*p* < 0.05). In contrast, groups GE30 and GE40 exhibited lower mRNA expression compared to GE20 without a significant difference from the control.

### 3.2. Effects on Oxidative Stress-Related Genes

The effects of GE on oxidative stress-related genes are shown in Figure 2. Oxidative stress-related genes *sod1* and *sod2* are significantly (*p* < 0.05) up-regulated in all experimental diets compared to the control group. The *gpx1* gene was significantly (*p* < 0.05) up-regulated only in the GE20 group, while the *cat* gene only showed a significant change in the GE40 group, indicating significant (*p* < 0.05) down-regulation at a higher amount of GE.

### 3.3. Effects on Thermal Tolerance

The critical minimum temperature after 60 days of GE feeding is shown in Figure 3. All experimental groups showed significantly (*p* < 0.05) higher cold tolerance as compared to the control group. The GE20 group exhibited the highest cold tolerance (mean 13.2 °C) compared to the Control, GE30, and GE40 (mean 14.03 °C, 13.6 °C, and 13.5 °C, respectively) groups.

## 4. Discussion

To our knowledge, no experiment has been conducted before to examine the effects of grape extract (GE) on oxidative stress-related enzyme regulations and evaluation of critical minimum temperature (CT_min_) in Asian seabass. Grapes are rich in polyphenols [10] and are easily available in various forms, such as grape seed, grape skin, and grape pomace flour, making them a popular choice for extracts to enhance the health of fish [35]. Other plants, such as olive extract, chestnut wood extract, and polygonum extract, are also sources of polyphenols used to improve the antioxidant capacity, immunity, and growth rate of Asian seabass [36,37]. This experiment was the first to investigate the effects of GE on growth performances and oxidative stress-related gene expressions in Asian Seabass and to evaluate their capability to increase cold stress tolerance after 60 days of feed supplement.

### 4.1. Antioxidant Enzyme-Related Genes Expression

Grapes (*Vitis vinifera*) are abundant in polyphenols such as catechin, epicatechins, procyanidins, resveratrol, tannins, and anthocyanins, recognized for their potent antioxidant properties [38]. The antioxidant activity of polyphenols is considered crucial in mitigating oxidative stress-induced damage [39]. Numerous internal defense mechanisms exist to counteract the harmful effects of free radicals. These compounds neutralize free radicals by donating electrons or hydrogen atoms, thereby preventing oxidative damage [40]. Additionally, they chelate metal ions like iron and copper, which are catalysts in forming reactive oxygen species (ROS), further reducing the oxidative burden [41]. Furthermore, polyphenols activate critical signaling pathways, including PI3K, Akt, and Nrf2, and in response to this, activate *gpx*, *sod*, *cat*, and *nadph*, which collectively detoxify harmful compounds like hydrogen peroxide [42,43].

*Sod1* is mostly found in the cytosol, while *sod2* and *sod3* are most abundant in the mitochondria and the extracellular space, respectively [44,45]. This study revealed a consistent increase in liver *sod1* and *sod2* enzyme activities across the GE20, GE30, and GE40 groups receiving GE compared to the control group, highlighting the improvements in antioxidant defenses similar to other studies in rainbow trout, common carp, greenlip abalone, rohu, and tilapia [10,13,46,47,48]. The GE20 group led to a significant increase in liver *gpx1* activity, establishing a positive correlation between GE concentration and *gpx1* expression, but higher doses in the GE30 and GE40 groups did not show any significant difference. Surprisingly, a higher dose of GE in the GE40 group resulted in reduced *cat* gene expression and showed a significant difference from all other groups. The *gpx1* enzyme plays a key role in neutralizing harmful ROS using *gsh*. The observed higher upregulation of *gpx1* compared to *cat* in response to *sod1* and *sod2* suggested a potential compensatory mechanism. While *cat*, located in peroxisomes, addresses H_2_O_2_, *gpx1* is active in mitochondria and becomes prominent, emphasizing a nuanced regulatory balance in mitigating oxidative stress [49]. These findings, similar to those of the research on common carp fed with grape seed proanthocyanidins (GSPE) up to 1500 mg/kg, revealed a comparable downregulation in *cat* activity as the supplementation level increased [50].

It suggests a complex relationship between GE dosage and its effects on antioxidant mechanisms, warranting further investigation into the broader impacts of GE on antioxidant pathways. It’s important to consider that various extraction methods and grape varieties may influence enzymatic activities differently due to the diverse composition of polyphenols present in the extracts of different grape species. Additionally, it was found that the concentrations required to induce *sod* activity were significantly higher than those needed for inducing *cat* activity, highlighting the intricate interplay between polyphenol concentration and enzymatic activities [50,51].

### 4.2. Growth Performances

In this study, the addition of GE to the diet influenced growth performance, with the GE20 group demonstrating superior results compared to the GE30 and GE40 groups across all growth parameters. GE contains polyphenol compounds known for their potential to enhance intestinal health, promote beneficial bacteria, and create a healthier gut environment; in response, the intestinal barrier is strengthened, and the body can better absorb essential nutrients, further supporting growth and overall health. The antimicrobial properties of these phenolic compounds not only contribute to a balanced gut microbiome but also prevent pathogenic bacteria from thriving, thus maintaining a healthy gastrointestinal tract. Additionally, they can facilitate the uptake of essential nutrients by chelating metal ions, which are crucial as cofactors for various enzymatic processes. These findings align with similar observations in other species, such as rainbow trout [13,52,53,54,55,56]. Similarly, the utilization of dietary crude GE also demonstrated improved weight gain, SGR, and condition factor in tambaqui juveniles [11].

Phytochemicals in GE, such as quercetin, rutin, tannins, and proanthocyanidins, also demonstrated significant enhancement of the growth-related gene *igf-1* in beluga sturgeon, tilapia, and grass carp [48,57,58]. The current study found that the GE20 group had significantly higher *igf-1* mRNA expression levels and weight gain compared to the GE30, GE40, and control groups. In other words, higher doses of dietary GE, exceeding 20 g/kg, could negatively impact the growth performance of Asian seabass. The reduced weight gain in fish fed higher levels of GE may be attributed to the negative effects of polyphenols at higher concentrations. Polyphenols, particularly condensed tannins, bind strongly to proteins, decreasing their digestibility and limiting the availability of essential amino acids [59]. This reduction in nutrient absorption has been observed in other animals, such as chickens and fish, where polyphenol-rich diets led to decreased growth performance due to impaired protein and amino acid utilization [10,60]. Additionally, polyphenols can interfere with the absorption of trace minerals, such as iron and zinc. Grape seed extract, in particular, has been shown to reduce zinc and heme iron absorption in a dose-dependent manner [61,62], further limiting growth. Therefore, as GE levels increase in the diet, the uptake of these crucial nutrients may be impaired, limiting overall growth. These findings suggest that an optimal dietary supplement level for promoting fish growth and health is likely necessary. While grape polyphenols offer potential benefits for gut health at moderate levels, excessive intake of proanthocyanidins and tannins in GE can reduce protein digestibility, lower plasma iron levels, and impair overall growth performance [63,64].

### 4.3. Thermal Tolerance

Cold shock, characterized by a sudden decrease in water temperature, poses significant threats to fish, leading to physiological, behavioral, and fitness consequences that can ultimately result in death [65]. Low temperature leads to increased production of ROS, imposing greater oxidative pressure that overwhelms the antioxidant defense system, thereby diminishing the enzyme’s efficacy [66].

The liver, gills, and brain, essential for metabolism, gas exchange, and neurological functions in fish, exhibit distinct responses to low temperatures. They activate specific antioxidative mechanisms to mitigate the rise in ROS [67,68]. Researchers have explored the use of antioxidants to enhance cold tolerance and mitigate the damage caused by oxidative stress in fish. Compounds like Vitamin E, known for their antioxidant properties, when added to the fish’s diet, have shown promise in reducing cold-induced oxidative stress and protecting tissues from damage, thereby improving their ability to withstand acute cold stress [69]. In vivo, the introduction of Vitamin C in zebrafish liver cells demonstrated the ability to alleviate H_2_O_2_ induced oxidative stress, consequently restoring cold tolerance [70]. Polyphenols play a crucial role in regenerating vitamin C oxidized form, dehydroascorbate, back to its active state by donating electrons or hydrogen atoms, ensuring its continued availability to combat oxidative stress. This interaction enhances the antioxidant defense, making polyphenols vital in supporting cold tolerance and protecting fish from oxidative damage [71].

Expanding on this concept, the study by Wang et al. (2020) [26] reported that resveratrol, an antioxidant found in grapes, enhances cold tolerance in tilapia by upregulating antioxidant genes such as *sod1*, *sod2*, and *gpx*. Polyphenols present in grapes modulate *nrf2*, releasing it from Kelch-like ECH-associated protein 1 in the cytoplasm. The *nrf2* then translocates to the nucleus, binds to the Antioxidant Response Element present in the promotor region of the DNA sequence, and induces antioxidant enzyme expression, neutralizing ROS and preventing cellular damage [39,72]. It is reasonable to assume that the grape extract (GE) used in this study contained resveratrol and other potent polyphenols with known antioxidant properties, which upregulated oxidative stress-related gene expressions, including *sod1*, *sod2*, and *gpx1*, after a 60-day feeding trial. These polyphenols have been shown to enhance cold tolerance by increasing the expression of specific genes. This suggests that polyphenols may play a crucial role in helping fish resist low temperatures and improve their survival chances during sudden temperature drops, such as cold snaps. Additionally, research in zebrafish and killifish has demonstrated that acclimation to low temperatures enhances mitochondrial respiration, maintains mitochondrial membrane potential, regulates ROS production during acute thermal shifts, and improves long-term cold stress tolerance [73,74]. These findings underscore the critical role of polyphenols in temporarily boosting cold resistance, potentially aiding fish in returning to normal conditions when given the opportunity.

## 5. Conclusions

This study investigates the impact of grape extract (GE) on the growth performance, oxidative stress regulation, and thermal tolerance of Asian seabass. Rich in potent polyphenols, GE is a strong antioxidant, mitigating oxidative stress through various mechanisms. The GE20 group exhibited the most significant results, notably increasing the expression of antioxidant enzymes-related genes such as *sod1*, *sod2*, and *gpx1*. This upregulation may correlate with enhanced *igf-1* expression and improved weight gain. The higher *gpx1* activity compared to *cat* suggests a compensatory response to the elevated antioxidant enzyme levels. Additionally, all GE-treated groups demonstrated higher cold tolerance compared to the control group. These findings offer valuable insights for the aquaculture industry, highlighting the multifaceted benefits of incorporating GE as a dietary supplement for Asian seabass.

## Figures and Tables

**Figure 1 animals-14-02731-f001:**
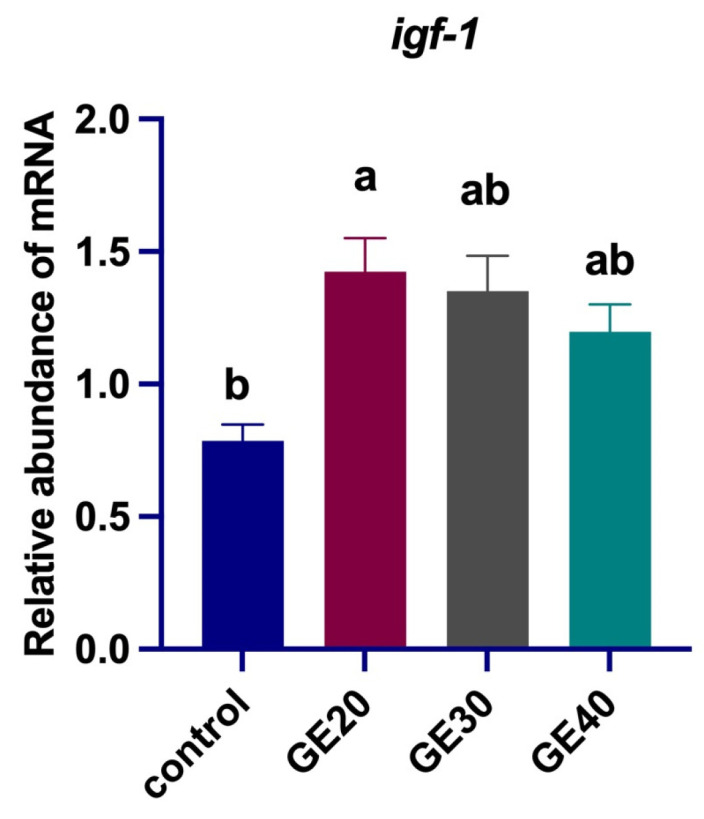
Comparisons of different levels of GE20, 20 g/kg; GE30, 30 g/kg; GE40, 40 g/kg of grape extract supplemented diets effect on the expression of growth-related gene Insulin-like growth factor 1 relative to *elf1α* in the liver of *Lates calcarifer*. Different letters indicate significant differences among groups by one-way ANOVA, and Duncan test values are mean ± SEM, *n* = 9 (*p* < 0.05).

**Figure 2 animals-14-02731-f002:**
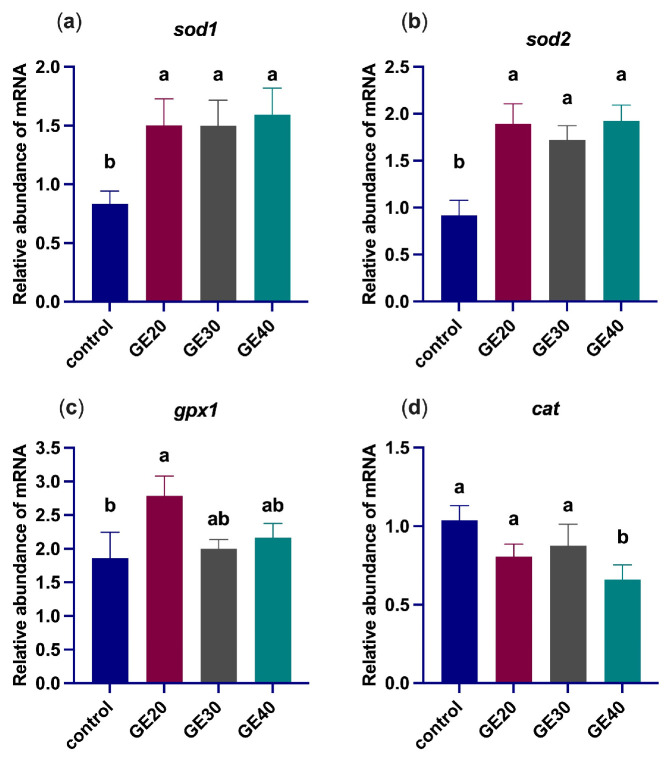
Comparisons of different levels of GE20, 20 g/kg; GE30, 30 g/kg; GE40, 40 g/kg of grape extract supplemented diets effects on the expression of antioxidant-related genes (**a**) superoxide dismutase 1, (**b**) superoxide dismutase 2, (**c**) glutathione peroxidase1, (**d**) catalase relative to *elf1α* in the liver of *Lates calcarifer*. Different letters indicate significant differences among groups by one-way ANOVA, and Duncan test values are mean ± SEM, *n* = 6 (*p* < 0.05).

**Figure 3 animals-14-02731-f003:**
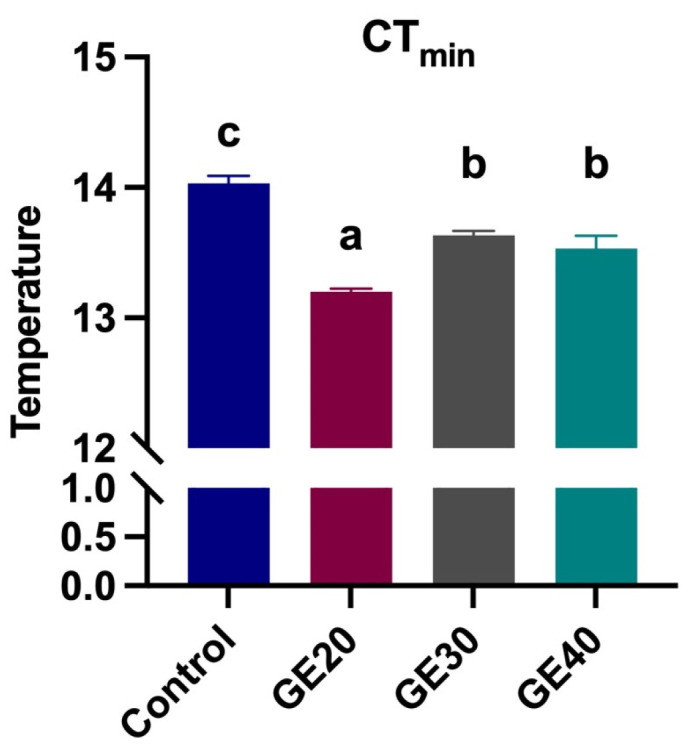
Comparisons of different levels of grape extract supplemented diets on critical thermal minimum (CT_min_) in *Lates calcarifer*. GE20, 20 g/kg; GE30, 30 g/kg; GE40, 40 g/kg. Different letters indicate significant differences among groups by one-way ANOVA, and Duncan test values are mean ± SEM, *n* = 9 (*p* < 0.05).

**Table 1 animals-14-02731-t001:** Primers detail used in qPCR.

Gene		Primer Sequence (5′ to 3′)	Amplicon Size (bp)	Reference Number
*igf-1*	F	ACGAGTGCTGCTTCCAAAG	118	XM_018697285.1
	R	GGTGTTCTCGGCATGTCTG		
*sod1*	F	GGTCCCAATGATGCAGAGAG	108	XM_018691152.1
	R	GGTCCCAATGATGCAGAGAG		
*sod2*	F	TGCGGCCAGACTATGTTAAG	200	XM_018675982.1
	R	GTATCAGTGTTGGTGGTCAGT		
*gpx1*	F	GGCTGGGAGTGTTGAAGAG	177	XM_018686718.2
	R	TTGCTGGAGTAACGAGAGTG		
*cat*	F	GAGTCTGCATCAGGTGTCTTT	109	XM_018675907.1
	R	CAAACTGGTTAATGCTGATGGG		
*elf1α*	F	GTTGCCTTTGTCCCCATCTC	130	XM_018699049.1
	R	CTTCCAGCAGTGTGGTTCCA		

The abbreviations used are as follows: *igf-1*, Insulin-like growth factor 1; *sod1*, superoxide dismutase 1; *sod2* superoxide dismutase 2; *gpx1*, Glutathione peroxidase1; *cat*, Catalase; *elf1α,* Elongation factor 1-alpha.

**Table 2 animals-14-02731-t002:** Growth performance and feed utilization of Asian seabass (*Lates calcarifer*).

Groups	Control	GE20	GE30	GE40
Initial weight (g)	20.1 ± 0.75	19.83 ± 0.75	20.33 ± 0.816	20.33 ± 1.032
Final weight (g)	80 ± 3.03 ^b^	93.5 ± 3.67 ^a^	81.3 ± 4.08 ^b^	81.66 ± 3.26 ^b^
Weight gain (g)	59.8 ± 3.43 ^b^	73.6 ± 4.17 ^a^	61 ± 3.74 ^b^	61.33 ± 3.26 ^b^
SGR% (g/day)	4.33 ± 0.038 ^b^	4.48 ± 0.039 ^a^	4.34 ± 0.04 ^b^	4.35 ± 0.04 ^b^
FI (%/day)	1.94 ± 0.20	1.92 ± 0.10	1.91 ± 0.19	1.96 ± 0.24
FCR	1.09 ± 0.04 ^a^	0.88 ± 0.04 ^c^	0.92 ± 0.06 ^ab^	0.96 ± 0.09 ^b^

FCR, feed conversion ratio; FI, feed intake; SGR, specific growth rate percentage; GE20, 20 g/kg; GE30, 30 g/kg; GE40, 40 g/kg. Different letters indicated significant differences within the group; values are means ± *SD*, *n* = 6 (*p* < 0.05).

## Data Availability

Data may be provided upon request to the corresponding author when the reason is justified.

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
