# Peer review of "Enhancement of Thermal Tolerance and Growth Performances of Asian Seabass (*Lates calcarifer*) Fed with Grape Extract Supplemented Feed"

_animals, 2024, doi:10.3390/ani14182731_

Round 1

Reviewer 1 Report

Comments and Suggestions for Authors

Enhancement of Thermal Tolerance and Growth of Asian Seabass  (Lates calcarifer) with Grape Extract Feed Supplementation

Review Report

The research topic in this manuscript is time demanding and has scientific merit for publication in ‘Animals.’ It is well written However, some of my specific suggestions are as below:

1.     The title can be changed to ‘’ Enhancement of Thermal Tolerance and Growth Performances of Asian Seabass (Lates calcarifer) fed with Grape Extract supplemented Feed.

2.     In the abstract grape extract (GE) supplementation has been expressed as g/kg feed; however, in methodology it has been expressed as percentage.

3.     Composition of GE need to be provided. 

4.     Confirm whether it is grape extract or grape seed extract.

5.     Table 2: Explanation needed, why weight gain increased in GE20; however, decrease in GE30 and GE40. Add unit of initial weight, final weight, and weight gain. Provide correct unit of SGR.

6.     Lines 268-269: compare with control.

7.     Lines 425, 426-427, and 460: incomplete references.

8.     Thoroughly check the reference check section.  

Comments on the Quality of English Language

English language need to be improved. 

Author Response

Reviewer #1:

Comment 1: The title can be changed to ‘’ Enhancement of Thermal Tolerance and Growth Performances of Asian Seabass (Lates calcarifer) fed with Grape Extract supplemented Feed.

Author’s Response:

Thank you for your valuable suggestion. We have revised the title to "Enhancement of Thermal Tolerance and Growth Performances of Asian Seabass (Lates calcarifer) Fed with Grape Extract Supplemented Feed" as recommended.

We believe this change more accurately reflects the focus of our study and appreciate your input.

Comment 2: In the abstract grape extract (GE) supplementation has been expressed as g/kg feed; however, in methodology it has been expressed as percentage.

Author’s Response:

We appreciate the reviewer’s observation regarding the discrepancy in the expression of grape extract (GE) supplementation. We have revised the abstract to align with the methodology, at line 27 expressing the GE supplementation as a percentage to ensure consistency in the manuscript. The text has been revised as “three diets supplemented with GE at 2% (GE20), 3% (GE30), and 4% (GE40) with commercial feed.”

Thank you for pointing this out, and we believe this change enhances the clarity of our presentation.

Comment 3: Composition of GE need to be provided. 

Author’s Response:

We appreciate the reviewer’s insightful question regarding the composition of the grape extract. In addition to the feed composition (protein, lipid content, etc) that has already been listed in the text, we have obtained a certificate from the supplier confirming that the grape extract contains 10% Resveratrol, as verified by HPLC. However, the company did not provide additional specific details about other polyphenols, presumably because resveratrol is one of the major constituents of public interest. While we also agree that knowing antioxidant compounds in GE may be beneficial to more-in-depth discussion, it is not the main focus of this investigation.  The identification of specific compounds (ex. polyphenol profiling) by HPLC will be under further investigation once the effect of GE in fish is proven (published). We hope this clarification addresses the concern.

Comment 4: Confirm whether it is grape extract or grape seed extract.

Author’s Response:

Thank you for pointing this out. To clarify, we have specified that the grape extract used in our study is derived from the “fruit” of Vitis vinifera at line 117 and 129. We have updated the manuscript to reflect this distinction clearly. We appreciate your attention to this detail.

Comment 5: Table 2: Explanation needed, why weight gain increased in GE20; however, decrease in GE30 and GE40. Add unit of initial weight, final weight, and weight gain. Provide correct unit of SGR.

Author’s Response:

Thank you for your valuable feedback. In response, we have made the following changes:

  1. Table 2 has been updated to include the correct units for initial weight (g), final weight (g), and weight gain (g) for each group to ensure clarity.
  2. We have also corrected the unit for the Specific Growth Rate (SGR) to properly reflect the correct formula (as % per day).
  3. To address the observed trend where weight gain increased in the GE20 group but decreased in the GE30 and GE40 groups, we have expanded the discussion from lines 348-362. This now includes an explanation of how the higher concentrations of polyphenols, particularly in the GE30 and GE40 diets, could have contributed to the reduced weight gain. As polyphenols are known to bind with proteins and reduce their digestibility, higher levels may have led to nutrient absorption inefficiencies and overall reduced growth performance, consistent with findings in previous studies. We have also elaborated on the negative effects of elevated polyphenol concentrations on the absorption of essential minerals such as iron and zinc, which may further explain the reduced growth in the higher GE groups.

Comment 6: Lines 268-269: compare with control.

Author’s Response:

Thank you for identifying this oversight. We have revised the text at line 273 to include a comparison with the control group, as suggested. We appreciate your attention to this detail and believe this revision improves the clarity of our findings.

Comment 7: Lines 425, 426-427, and 460: incomplete references.

Author’s Response:

Thank you for pointing out the issues with the references. We have corrected the references at lines 434-436 and have reviewed all other references to ensure consistency.

  1. Manam, V.K. Fish feed nutrition and its management in aquaculture. J. Fish. Aquat. Stud. 2023, 11, 58-61.
  2. Ayala-Zavala, J.F.; González-Aguilar, G.; Siddiqui, M.W. Plant food by-products: industrial relevance for food additives and nutraceuticals; CRC Press: 2018.

Reviewer 2 Report

Comments and Suggestions for Authors

This article is well written and the main idea is to expose the effects on growth and genes involved in antioxidant functions in Asian Seabass, due to the inclusion of grape extract in feed.

I have detected the following mistakes:

Line 77 --> a full stop is missing after ‘low temperatures’. 

line 307 and 308 --> the text says: ‘However, the CAT enzyme activity showed no statistically significant difference in response to the GE dosage’. On the contrary, the graphs show that the higher GE40 dosage significantly reduces catalase levels, which is mentioned on line 313.

Line 314 --> GE30 and GE40 do not reduce the levels of gpx1, as they are similar to those obtained in the controls, so I suggest to modify the sentence mentioning that only the GE20 group was able to significantly increase the levels of gpx1 with respect to those obtained in the controls.

For these last two errors (line 307/308 and 314), I suggest that the results be rewritten in a way that clarifies the concepts and agrees with the graphs presented.

I would like to include in the results the levels of feed intake by each group per day and the cumulative intake throughout the experiment. This suggestion is made because the animals in the GE20 group grow significantly more than the other groups, however the FCR is not different with respect to those in the GE20 group.

Bibliographic references are adequate, although I recommend that it be reviewed exhaustively and written the same in all citations. Some citations include the doi and others do not, in some citations the doi is presented in web format and others not, in some of them the doi contains a hyperlink and I suggest that it be deleted. In summary I suggest that all references be written in the same way.

Author Response

Reviewer #2:

Comment 1: Line 77 --> a full stop is missing after ‘low temperatures’. 

Author’s Response:

Thank you for pointing out this punctuation error. We have corrected the issue by adding the full stop after 'low temperatures' in line 78. We appreciate your attention to detail.

Comment 2: line 307 and 308 --> the text says: ‘However, the CAT enzyme activity showed no statistically significant difference in response to the GE dosage’. On the contrary, the graphs show that the higher GE40 dosage significantly reduces catalase levels, which is mentioned on line 313.

Author’s Response:

Thank you for highlighting this discrepancy. We have revised the text to accurately reflect the data shown in the graphs. The paragraph has been corrected at lines 311 and 313 to align with the observed significant reduction in catalase levels with the GE40 dosage. We appreciate your attention to this inconsistency and believe this revision clarifies the results.

Comment 3: Line 314 --> GE30 and GE40 do not reduce the levels of gpx1, as they are similar to those obtained in the controls, so I suggest to modify the sentence mentioning that only the GE20 group was able to significantly increase the levels of gpx1 with respect to those obtained in the controls.

Author’s Response:

Thank you for your helpful suggestion. We have revised the text at lines 310 and 311 to accurately reflect that only the GE20 group significantly increased GPx1 levels compared to the controls, while the levels in GE30 and GE40 groups remained similar to those of the control group. We appreciate your guidance in improving the accuracy of our findings.

Comment 4: For these last two errors (line 307/308 and 314), I suggest that the results be rewritten in a way that clarifies the concepts and agrees with the graphs presented.

Author’s Response:

Thank you for your suggestion. We have reviewed the results section and ensured that the text accurately reflects the data shown in the graphs. We appreciate your feedback, which has helped enhance the clarity of our findings.

Comment 5: I would like to include in the results the levels of feed intake by each group per day and the cumulative intake throughout the experiment. This suggestion is made because the animals in the GE20 group grow significantly more than the other groups, however the FCR is not different with respect to those in the GE20 group.

Author’s Response:

We appreciate your insightful suggestion. In response, we have incorporated the feed intake % per day for each group into Table 2 and have added the relevant formula in line 168-169. This addition helps to clarify the relationship between feed intake and growth performance and addresses the feed conversion ratio (FCR) observations. Thank you for helping us improve the clarity and completeness of our results.

Comment 6: Bibliographic references are adequate, although I recommend that it be reviewed exhaustively and written the same in all citations. Some citations include the doi and others do not, in some citations the doi is presented in web format and others not, in some of them the doi contains a hyperlink and I suggest that it be deleted. In summary I suggest that all references be written in the same way.

Author’s Response:

Thank you for your detailed feedback on the references. We have thoroughly reviewed and revised all citations to ensure consistent formatting. We have deleted the DOI numbers to maintain uniformity throughout the reference list. We appreciate your careful review and suggestions, which have helped us enhance the clarity and consistency of our references.
